# Guava (*Psidium guajava* L.) Fruit and Valorization of Industrialization By-Products

Jorge E. Angulo-López [1], Adriana C. Flores-Gallegos [1,*], Cristian Torres-León [2], Karen N. Ramírez-Guzmán [3], Gloria A. Martínez [1] and Cristóbal N. Aguilar [1,*]

1   Department of Food Research, Faculty of Chemical Sciences, Universidad Autónoma de Coahuila, Saltillo 25280, Mexico; lopez.eduardo@uadec.edu.mx (J.E.A.-L.); alicia.martinez@uadec.edu.mx (G.A.M.)
2   Research Center and Ethnobiological Garden, Universidad Autónoma de Coahuila, Viesca 27480, Mexico; ctorresleon@uadec.edu.mx
3   Center for Interdisciplinary Studies and Research, Universidad Autónoma de Coahuila, Saltillo 25016, Mexico; nathiely_ramirez@uadec.edu.mx
*   Correspondence: carolinaflores@uadec.edu.mx (A.C.F.-G.); cristobal.aguilar@uadec.edu.mx (C.N.A.)

**Abstract:** Guava (*Psidium guajava* L.), is a fruit considered native to the American tropics. It is of great economic importance in many countries of the world, due to its high production and diversity of products derived from its fruit. It can be consumed fresh or processed. During the fruit's processing, some residues are generated, such as seeds, rinds, and pulp, which is about 30% of the fresh fruit's volume. Different studies have shown that it can be used in various industries, such as food, chemical, and pharmaceutical. By-products or processing residues have valuable components. Evidence indicates that it contains significant amounts of dietary fiber (soluble and insoluble), vitamins (A, B, C, β-carotene), essential oils, minerals, proteins (transferrin, ceruloplasmin, albumin), pectins, antioxidants (flavonoids, flavonols, condensed tannins) and volatile organic compounds; these elements can help in the prevention of chronic and degenerative diseases. *P. guajava* L. is considered a nutraceutical due to its compounds with beneficial properties on health and disease prevention. Therefore, this paper aims to review the physicochemical composition of the different by-products of the processing of guava and its reported uses.

**Keywords:** guava; *Psidium guajava*; waste; nutraceuticals; by-products

## 1. Introduction

Guava (*Psidium guajava* L.) is a fruit native to the American tropics; it is also known as guayabo, guara, arrayana, and luma in some regions [1]. It ranges from Mexico to Peru, but because of its adaptability, it is cultivated in tropical and subtropical zones of Europe, Africa, and Asia [1,2]. This fruit was carried across the Pacific by the Spanish to the Philippines, and by the Portuguese to India [2], and it was rapidly adopted as a crop in Asia and some areas of Africa; it is now found in Egypt, Palestine, Algeria, and the French Mediterranean coast [2]. It is grown in almost all tropical and subtropical countries, at elevations ranging from 0 to 2000 meters (m), in a wide range of climates and soils, with annual rainfall ranging from 1000 to 2000 mm and average temperatures ranging from 20 to 30 °C [3].

India, Brazil, the Philippines, Mexico, Colombia, Peru, Ecuador, South Africa, the United States of America, Venezuela, Costa Rica, Cuba, and Puerto Rico are the primary producers of this fruit. The majority of the varieties sold in Europe are imported from South Africa and Brazil. Commercially, they are classified as white or red depending on the color of the pulp [3].

Guava is considered a super fruit by some researchers due to the high content of phenols and other antioxidant substances [4]. Fresh guava trade is limited internationally, but processed guava products, such as preserves and drinks, are becoming more common

in many countries [5]. Guava is commonly used in the food industry to produce pulp, nectars, jams, jellies, and syrups [6]. The residues or by-products, mostly seeds, account for approximately 30% of the weight of the processed fresh fruit. Because of its sensory properties and the presence of bioactive compounds, this fruit and its by-products have the potential to be integrated into healthy processed foods [4]. However, these by-products are often discarded in landfills without being treated, causing environmental problems [6–8]. For example, in Brazil, one of the major producing countries, more than 70 thousand tons of waste are discarded, despite the fact that it contains a high concentration of bioactive compounds [4]. The valorization of agroindustrial waste has emerged as an appealing option for using them and creating new goods while minimizing the emissions they represent [9].

By-products of fruit and vegetable processing are the most studied substrates for the extraction of different antioxidants and dietary fibers [10]. Guava by-products have a higher total dietary fiber content than cereals and pseudocereals like oats (*Avena sativa*), barley (*Hordeum vulgare*), rye (*Secale cereale*), quinoa (*Chenopodium quinoa* Willd.), amaranth (*Amaranthus caudatus*), and chia (*Salvia hispanica* L.) [8].

### 1.1. Taxonomic Hierarchy

The classification that follows refers to the information from the Integrated Taxonomic Information System: [11,12]

| | |
|---|---|
| Kingdom | Plantae |
| Subkingdom | Viridiplantae |
| Infrakingdom | Streptophyta |
| Superdivision | Embryophyta |
| Division | Traqueofita |
| Subdivision | Spermatophytina |
| Class | Magnoliopsida |
| Superorder | Rosanae |
| Order | Myrtales |
| Family | Myrtaceae |
| Genus | *Psidium* |
| Species | *P. guajava* |

### 1.2. Botanical Description

Guava is a plant in the Myrtaceae family [3,11], which includes about 133 genera and 3800 tree and shrub species. The genus *Psidium* contains about 150 species, the most notable are *P. cattleianum* Sabine, *P. fredrichsthalianum* (Berg) Nied, and *P. guajava* L. Theseare known for its economic and commercial importance [3,13], the high nutritional value of its fruits, which are high in vitamins A, B, and C, the numerous medicinal applications for which its fruits, leaves, flowers, bark roots, and stems are used [14], as well as the profitability of its cultivation [13].

*P. cattleianum*, known as arazá, is a native Brazilian fruit [15,16]. Its fruit is small, slightly round in shape, 2–3 cm in diameter, with thin violet-red skin; its pulp is soft, white, and juicy, with a sweet, acidic, and spicy taste. It has a lot of tiny, white-colored seeds [2,15].

*P. friedrichsthalianum*, known as "cas" or "Costa Rican guava" in Costa Rica, is native to Honduras and is cultivated from northern South America to southern Mexico [16]. The fruit is rounded in shape, slightly flattened at the ends, with a length of 3.56 cm and a diameter of 4.29 cm, yellow skin, and mildly acidic pulp; it is used to make candy, juices, and jellies [2,16].

*P. guajava* L. is the most resistant tropical fruit tree due to its adaptability and high production rate [17]. It exhibits a high level of variability in populations, with distinct fruit sizes, pulp and peels color, seed number, and other morphological characteristics [14].

Depending on the climatic conditions and variety, bushes or trees up to 10 m tall with a short and twisted stem can be found [11]. Nonetheless, if pruned correctly, it does

not exceed three meters [13] of bark in the form of brown scales. It can produce roots up to five meters deep, depending on the type of soil and water table, giving it excellent anchorage [18]. The tree grows quickly, bears fruit in two to four years, and continues to bear fruit for another 40 to 60 years [17]. Flowering times differ depending on the environment in which it is grown. Guava trees flower and bear fruit all year in temperate and tropical temperatures [2]. Its flowers are hermaphrodites, 1.5 to 2 cm long and 3.8 cm wide, axillary, and pedicellate, with actinomorphic epigenetics [18] and four to five white petals [11] (Figure 1).

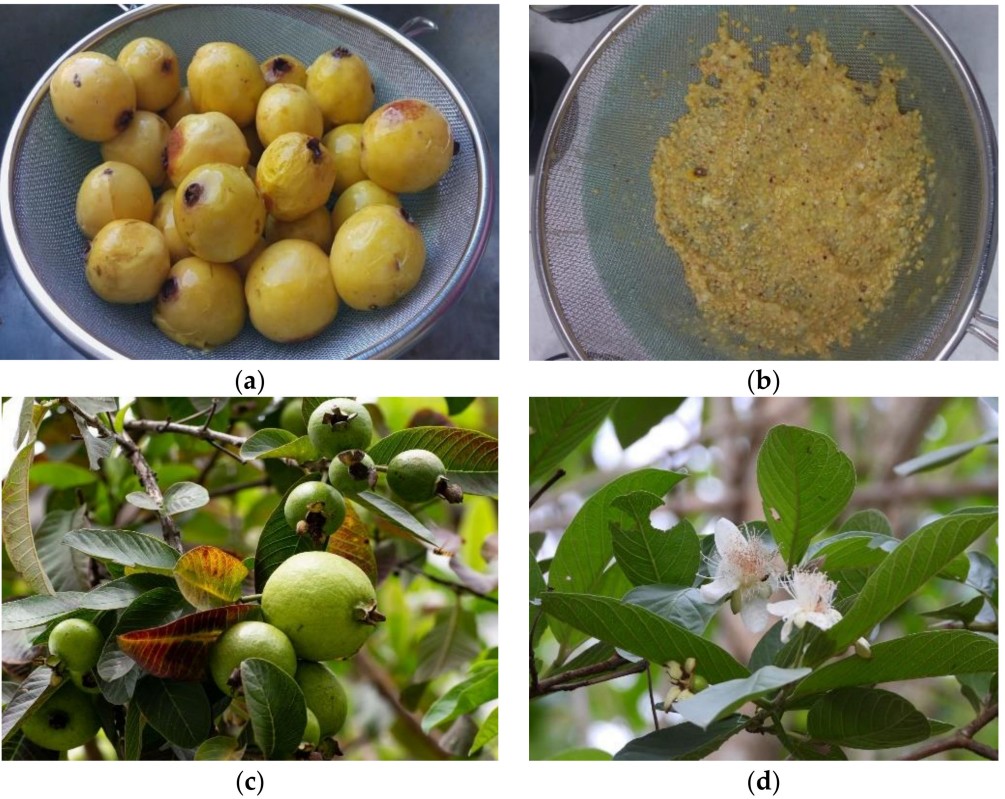

**Figure 1.** Description of guava plant (*P. guajava* L): (**a**) Fruit, (**b**) waste or by-products, (**c**) guava plant, and (**d**) flower and leaves.

## 2. National and International Production

Guava cultivation is of great economic significance in many countries around the world, owing to its high yield and the variety of products derived from its fruit [19–21]. Guava is grown in over 60 countries [21], and its worldwide production is estimated to be around 40 million tons (2020) [17]. India is the world's biggest guava producer, followed by China and Kenya [2], with Brazil and Venezuela also standing out [21]. Mexico ranks fifth in the world with a production of 302,718 tons per year (2017), and an increase is expected due to a 5% increase in the harvested area; 4% of total production is exported, primarily to the United States and Canada [22]. Guava crops can be found almost everywhere in the country; there are commercial, wild plantations and family or backyard gardens; the most important producing areas are in the states of Michoacán, Aguascalientes, and Zacatecas, particularly in the "Calvillo-Caones" region [21,23].

## 3. Industrialization and Generation of By-Products

Agro-industry generates millions of tons of waste worldwide [24], which increases production costs in developed countries and has a serious environmental effect [9]. Although some waste is processed and used as a fiber source in animal feed or fields, the vast majority is discarded untreated [24,25]. Several studies have shown that these residues are high in bioactive compounds that can protect against oxidative damage caused by

free radicals. Fruit and vegetable peels are high in bioactive ingredients like fiber and antioxidants [24].

During guava processing, the pulp is obtained as the primary product, with seeds and peels obtained as by-products [26], which can account for up to 30% of the volume of the fresh fruit [11]. Table 1 illustrates the values reported in the various studies on the physicochemical characterization of guava and its by-products that were consulted. The values of the reference [f] correspond to values on a dry basis.

**Table 1.** Physicochemical composition of pulp and seeds of *Psidium guajava*.

| Proximate Analysis. | Pulp | Seeds | Reference |
|---|---|---|---|
| Moisture (%) | [b] 85;<br>[f] 6.41 ± 0.11;<br>[g] 84.9 | [a] 6.68 ± 0.00;<br>[c] 8.3 ± 0.03;<br>[d] 9.3 ± 0.03 | [a] [24], [b] [19], [c] [27], [d] [6], [f] [7], [g] [8] |
| Protein (g/100 g) | [b] 0.3<br>[f] 5.13 ± 0.26<br>[g] 0.88 | [a] 11.19<br>[d] 4.8 ± 0.10<br>[h] 7.71 | [a] [2], [b] [4], [d] [26], [f] [28], [g] [8], [h] [29] |
| Fiber | | | |
| Total dietary fiber (g/100 g) | [b] 2.4<br>[f] 43.21 ± 0.09 | [a] 63.94<br>[d] 69.1 ± 0.17<br>[h] 69.63 | [a] [24], [d] [26], [f] [28], [h] [29] |
| Insoluble dietary fiber (g/100 g) | [f] 42.56 ± 0.06 | [a] 63.55<br>[d] 57.7 ± 0.15 | [a] [24]<br>[d] [26], [f] [28] |
| Soluble dietary fiber (g/100 g) | [f] 0.65 ± 0.04 | [d] 11.1 ± 0.09 | [d] [26], [f] [28] |
| Ether Extract (g/100 g) | [b] 0.1<br>[f] 4.32 ± 0.24<br>[g] 0.53 | [d] 1.4 ± 0.10<br>[h] 10.12 | [b] [19], [d] [26], [f] [28], [g] [8], [h] [29] |
| Carbohydrates (g/100 g) | [b] 15<br>[g] 13.2 | [d] 22.2 ± 0.14<br>[h] 11.51 | [b] [19]; [d] [26], [g] [8], [h] [29] |
| Ashes (g/100 g) | [b] 0.5<br>[e] 0.52 ± 0.05<br>[f] 5.04 ± 0.39<br>[g] 0.43–0.7 | [a] 1.18 ± 0.02<br>[d] 2.4 ± 0.10<br>[e] 0.66 ± 0.04<br>[h] 1.01 | [a] [24], [b] [19], [d] [26], [e] [30], [f] [28], [g] [8], [h] [29] |
| Vitamins and minerals | | | |
| Vitamin A (IU/100 g) | [b] 109<br>[g] 200–400 | [h] 50.13 | [a] [24], [g] [8], [h] [29] |
| Thiamine (B1) (mg/100 g) | [b] 0.06<br>[g] 0.046 | | [b] [19], [g] [8] |
| Riboflavin (B2) (mg/100 g) | [b] 0.06 [g] 0.03–0.04 | | [b] [19], [g] [8] |
| Niacin (B3) (mg/100 g) | [b] 1.3<br>[g] 0.6–1.068 | [h] 0.16 | [b] [19], [g] [8], [h] [29] |
| Ascorbic acid (C) (mg/100 g) | [b] 190<br>[g] 100 | [a] 87.44<br>[h] 0.20 | [a] [24]<br>[b] [19], [g] [8], [h] [29] |
| Zinc<br>(mg/100 g) | | [a] 3.31 | [a] [2] |
| Calcium (mg/100 g) | [b] 15<br>[g] 9.1–17 | [c] 0.05 ± 0.14<br>[h] 60.07 | [b] [19], [c] [27], [g] [8], [h] [29] |
| Phosphorus (mg/100 g) | [b] 16<br>[g] 17.8–30 | [h] 160.55 | [b] [19], [g] [8], [h] [29] |
| Iron (mg/100 g) | [b] 0.3<br>[g] 0.30–0.70 | [a] 13.8<br>[h] 3.32 | [a] [24], [h] [29]<br>[b] [19], [g] [8] |
| Potassium (mg/100 g) | [b] 292 | [h] 300 | [b] [19], [h] [29] |
| Sodium (mg/100 g) | [b] 6 | | [b] [19] |
| Calories kcal /100 g | [b] 54.97<br>[g] 36–50 | [a] 182 | [a] [24], [b] [19], [g] [8], |

**Table 1.** *Cont.*

| Proximate Analysis. | Pulp | Seeds | Reference |
|---|---|---|---|
| Unsaturated fatty acids (%) | | [a] 87.06 | [a] [24] |
| Bioactive Compounds | | | |
| Ascorbic acid (mg/100 g) | | [a] 87.44 | [a] [24] |
| Total carotenoids (mg/100 g) | | [a] 1.25 | [a] [24] |
| Total phenols (mg GAE/g) | [f] 44.04 ± 0.56 | | [f] [28] |

### 3.1. Fruit

The fruit can be round or pear-shaped, and its weight can range from 25 to 500 g [18,19]. It is 4–12 cm long and 4–7 cm wide, and it is distinguished by its aromatic, soft, and sticky pulp. The color of the pulp varies greatly: it may be white, yellow, pink, orange, or salmon [11,19], while commercially they are classified as white and red based on the color of the pulp. The pulp can be dense with few seeds in the center or thin with multiple seeds being part of the pulp [19].

According to the Mexican Norm NMX-FF-040-1993, the guava is a "fruit with a globose, ovoid, or pyriform shape, yellow-green on the outside or light yellow at full maturity; the pulp is yellowish-white, pink, or red, with a sweet or acid and aromatic flavor; the seeds are abundant and yellow, belonging to the Myrtaceae family, genus *Psidium*, and species guajava. Guava production and consumption in Mexico is concentrated on the "Chinese media" and "Peruvian" varieties, which were picked from germplasm; it can be eaten fresh or processed into a variety of products [31].

Guava is commonly consumed in Latin America and around the world, either as fresh fruit or in products such as juices and candies. Aside from its pleasant flavor, the rising demand has been boosted by new lifestyle trends that encourage society to adopt healthy habits [32–35]. The fruit has low carbohydrate, fat, and protein content and high water content [1], as well as vitamins A, B, and C. The vitamin C content is between 180 and 300 mg per 100 g of fruit, which is far greater than that found in citrus fruits like oranges and lemons [34,35], which is why it is referred to as a "superfruit" [4,36]. Several studies regarded ripe guava as a nutraceutical due to its compounds with health and disease-prevention properties [1].

*P. guajava* L. is a fruit that is known around the world as a food, but it is also used as medicine by several indigenous peoples in Central America and Africa, and it is present in many traditional medicines. According to Gutiérrez et al. [1], various research and clinical studies have been developed to explain the specific bioactivity of individual phytochemicals extracted from guava.

Documented scientific research on the medicinal properties of guava dates back to the 1940s [1], and it has been shown that the guava plant, fruit, and processing residues contain significant amounts of essential oils, vitamins (A and B), calcium, iron, potassium, pectins, and antioxidant substances such as phenolic compounds, ascorbic acid, carotenoids, lycopene, volatile organic compounds, and elements. They have the potential to aid in the prevention of chronic and degenerative diseases such as cancer [3,4,17,37–39]. *P. guajava* L. has been shown in studies to have hepatoprotective, antiallergic, antimicrobial, antigenotoxic, antiplasmodial, cytotoxic, antispasmodic, cardioactive, antidiabetic, anti-inflammatory, and antinociceptive properties [1]. The compounds, particularly those extracted from the leaves and fruits, have beneficial pharmacological properties.

The chemical composition and concentration of these components differ dramatically depending on the species or variety, fruit maturity, cultivation conditions, soil type, environment, and agricultural practices [17]. Because of all of these compounds found not only in the fruit and by-products, but also in the plant, guava fruits are considered ethnopharmaceutical drugs, traditionally used to treat diarrhea, throat inflammation, and for their high antibacterial activity against *Salmonella*, *Serratia*, and *Staphylococcus* [34].

The phenolic compound content of powder of *P. guajava* L. ranges from 44 to 516 mg GAE/100 g [28,37]. Pink pulp has values ranging from 170 to 300 mg GAE/100 g [28]. Gallic acid, chlorogenic acid, ellagic acid, catechin, and rutin are the most common phenolic compounds contained in pulp [37]. The antioxidant activity of the extracts correlates with the presence of a wide range of phenolic compounds [2,37].

### 3.2. Leaves

The tree's leaves are leathery, oval, or oblong-elliptical in shape, with short, smooth, and light green to dark green petioles arranged in semi-alternating pairs [18], with a midrib and several secondary leaves that emphasize a plain perspective [13]. When smashed, the approximately 3–16 cm long and 3–6 cm wide oblong [2,11] presents a distinctive scent, which is characteristic of essential oil, and the smell depends on the cultivar [13].

The plant *P. guajava* L., which is widely used as an edible fruit, has long been used to treat a variety of gastrointestinal issues. The leaves of *P. guajava* L. have been used in traditional medicine in Taiwan, Japan, China, and Korea [40]. Numerous studies have shown that guava leaves have anti-hyperglycemic and anti-hyperlipidemic properties. *P. guajava* L. has been shown to have anti-diabetic activity in vitro by Khaleel and Kumari [41] and Díaz-de-Cerio et al. [42]. The authors proved that the methanol extract at a concentration of 50 g plant extract/L was more potent than all extracts, with the lowest mean glucose concentration of 201 + 1.69 mg/dl after 27 h. The methanolic extract inhibited glucose diffusion significantly in vitro, confirming the plant's traditional claim. Guava has also been used as a hypoglycemic agent in Taiwanese traditional medicine [43]. The presence of bioactive plant molecules such as phenolic compounds is attributed to the biological properties described in *P. guajava* L. Polyphenols, also known as phenolic compounds, are a type of secondary metabolite that has been used in preventive medicine for centuries [44]. *P. guajava* L. leaves have gotten a lot of attention because they contain more phenolic compounds than the rest of the tree [1] (Table 2).

**Table 2.** Total polyphenol content in *P. guajava* L. leaves.

| Country | Total Polyphenols (mg/g d.w) [a] | References |
|---------|---------------------------------|------------|
| México | 7.5 ± 0.3 | [45] |
| Taiwan | 414–483 | [43] |
| Taiwan | 261.2 | [46] |
| Spain | 157 ± 6 | [40] |
| Indonesia | 101.2–101.9 | [47] |
| China | 50.57 | [48] |

[a] Expressed as mg gallic acid equivalent (GAE), d.w: dry weight.

The *P. guajava* L. leaves have a high total phenolic content ranging from 7.5 to 483 mg/g dry weight. *P. guajava* L. leaves are also high in proanthocyanidins, which could be used in nutraceutical formulations [49].

Purification of *P. guajava* L. leaf extracts results in the isolation of quercetin, quercetin-3-arabinoside, and Asian acid with wide antimicrobial activity against bacteria, fungi, viruses, and parasites, capable of treating diarrhea, gastroenteritis, dental plaque, acne, childhood rotavirus enteritis, and even malaria, as well as antioxidant properties and an inhibitory effect on the frequency of cough. Other compounds in guava leaves, such as β-sitosterol, flavonoids, triterpenoids, and volatile oil, may explain some of the advantages in traditional and ethnomedical uses of the plant in the management or control of some diseases, particularly dermatological disorders [1].

Polyphenols are the principal phytochemical compounds found in guava leaves, pulp, and peels. Díaz-de-Cerio et al. [50] discovered 75 phenolic compounds in the leaves of *P. guajava* L. These researchers employed HPLC-DAD-ESI-QTOF-MS. Meanwhile, Rojas-Garbanzo et al. [51] used UHPLC-DAD-MS/MS to identify 61 polar compounds in the peel and pulp of *P. guajava* L. Table 3 shows the identification results: retention periods and

m/z values for the principal phenolic compounds identified by the authors. The phenolic compounds were extracted using an ultrasound bath and a mixture of ethanol/water (80/20, *v/v*) and methanol/water (9/1, *v/v*) for the leaves and peel/pulp, respectively.

The phenolic compounds found in the leaves of *P. guajava* L. belong to the flavonols (76%), flavan-3-ols (45%), gallic and ellagic acid derivatives (35%), and flavanones families (1%). Díaz-De-Cerio et al. [40] investigated the identification and quantification of polar compounds in guava leaf extracts (ultrasound aqueous extract and infusions).

According to the quantification findings, the polar compound concentrations in all samples were flavonols > flavan-3-ols > gallic and ellagic acid derivatives > benzophenones > flavanones. According to Amaya-Cruz et al. [45], the principal polyphenols in *P. guajava* L. leaves were p-hydroxybenzoic acid and epicatechin. According to Liu et al. [52], the extract contains $2.25 \pm 0.29\%$ catechin and $1.45 \pm 0.13\%$ epicatechin, respectively. According to Rojas-Garbanzo et al. [51], the phenolic compounds found in the pulp and peel of *P. guajava* L. belong to the family of ellagitannins, flavones, flavonols, proanthocyanidins, dihydrochalcones, and anthocyanidins, as well as non-flavonoids such as phenolic acid derivatives, stilbenes, etc. The principal compound in the peel was cinnamoyl-glucoside, while the pulp contained catechin-gallate. The antioxidant activity and phytochemical composition of *P. guajava* L., on the other hand, differ substantially depending on cultivar, growing conditions, and extraction method [53].

The most common technique is solvent extraction of solid–liquid mixtures. Because of good manufacturing practices, ethanol and water are the most widely used extraction solvents in food systems [54]. In the extraction of polyphenols from guava leaves, Díaz-de-Cerio et al. [49] tested pure ethanol and various hydroethanolic mixtures, such as ethanol/water ratios of 90:10, 80:20, 70:30, 60:40, and 50:50 (*v/v*). The mixture of EtOH/$H_2O$ 80:20 (*v/v*) had the most phenolic compounds. Polyphenols occur in nature as both soluble and insoluble-bound molecules. Polyphenols that are soluble are easier to remove than phenols that are insoluble (due to their interaction with proteins or polysaccharides in the cell wall). The selection of the suitable method for the release of all phenolic compounds is critical in order to achieve bioactive extracts with high antioxidant power. Infusions are used in conventional medicine to extract bioactive molecules; however, green technologies have piqued the science community's interest.

The primary new methods for extracting bioactive molecules from the guava plant are ultrasound-assisted extraction [52], supercritical fluid extraction [56], and microwave-assisted extraction [57]. Furthermore, biotechnological processes such as solid-state fermentation with microorganisms such as mushrooms and bacteria can be used [58]. Wang et al. [59] found that fermenting guava leaves with *Monascus* and *Bacillus* increased the release of bound polyphenolics. The antioxidant properties were substantially enhanced using the solid-state fermentation method.

Polyphenols are widely thought to have powerful biological properties [59]. *P. guajava* L. extracts were discovered to have high antioxidant activity. The antioxidant mechanisms of guava extract's bioactive components may be attributed to their ability to scavenge free radicals [43]. The guava fruit is high in antioxidants, and the leaf has far greater bioactivity than the fruit [43]. Total polyphenols extracted from guava leaf showed higher bioactivities in scavenging 1,1-Diphenyl-2-picrylhydrazyl (DPPH) and 2,2-Azino-bis 3-ethylbenzothiazoline-6-sulfonic acid diammonium salt (ABTS) caption radicals, with $IC_{50}$ values of 30.57 g/mL and 24.53 g/mL for DPPH and ABTS$^+$ radicals, respectively [48]. Guava leaf extracts' antioxidant properties have been studied as food preservatives to replace synthetic antioxidants. At 4000 ppm or greater, the guava leaf extract is effective in avoiding oxidation in fresh pork sausage [60]. The fruit of *P. guajava* L. has a high potential for valorization in the selection of strong bioactive compounds; as a result, their use can be recommended for application in polyphenol-based food products or medication with enhanced health advantages and antioxidant functions.

**Table 3.** Phenolic compounds identified in the peel, pulp, and leaves of *P. guajava* L.

| Peel and Pulp [a] | | |
|---|---|---|
| Compound | Rt (min) | m/z |
| Phenolic acid derivatives | | |
| Galloyl-hexoside | 1.8 | 331 |
| Galloyl-hexoside | 2.1 | 331 |
| Gallic acid | 3.01 | 169 |
| Galloyl-pentoside | 5.7 | 301 |
| Hydroxybenzoyl-galloylglucoside | 7.4 | 453 |
| Dimethoxycinnamoyl-hexoside | 11.25 | 415 |
| Dimethoxycinnamoyl-hexoside | 11.3 | 415 |
| Flavones | | |
| Chrysin-C-hexoside | 11.95 | 415 |
| Ellagitannins | | |
| Valoneic acid bilactone | 13.75 | 469 |
| Flavonols | | |
| Quercetin-galloyl-hexoside | 12.16 | 615 |
| Quercetin-hexoside | 12.3 | 463 |
| Quercetin-hexoside | 12.5 | 463 |
| Quercetin-glucuronide | 12.9 | 477 |
| Quercetin-pentoside | 13.39 | 433 |
| Quercetin-pentoside | 13.6 | 433 |
| Quercetin-pentoside | 13.92 | 433 |
| Quercetin-galloyl-pentoside (guavinoside C) | 16.67 | 585 |
| Quercetin-deoxyhexoside-hexoside | 17.95 | 609 |
| Quercetin | 18.63 | 301 |
| Monomeric flavanols | | |
| Epigallocatechin | 6.01 | 305 |
| Catechin | 6.55 | 289 |
| Epicatechin | 7.81 | 289 |
| Gallocatechin gallate | 9.5 | 457 |
| Epigallocatechin gallate | 10.53 | 457 |
| Catechin gallate | 11.63 | 441 |
| Epicatechin gallate | 13.47 | 441 |
| Proanthocyanidins | | |
| PAC B-Type (E)GCg-(E)GC | 2.71 | 913 |
| PAC B-Type (E)GC-(E)GC | 3.81 | 609 |
| PAC B-Type (E)GC-(E)GC | 4.64 | 609 |
| PAC B-Type (E)GC-(E)C | 4.93 | 593 |
| PAC B-Type (E)GC-(E)C | 5.03 | 593 |
| PAC B-Type (E)C-(E)GC | 5.48 | 593 |
| PAC B-Type (E)GC-(E)C | 5.97 | 593 |
| PAC B-Type (E)GC-(E)GC | 6.79 | 609 |
| PAC B-Type (E)C-(E)C | 6.8 | 577 |
| PAC B-Type (E)C-(E)GC-(E)GC | 7.07 | 897 |
| PAC B-Type (E)Cg-(E)C | 7.26 | 881 |
| PAC B-Type (E)C-(E)GC | 8.32 | 593 |
| PAC B-Type (E)GC-(E)C | 8.44 | 593 |
| PAC B-Type (E)C-(E)C-(E)C | 8.67 | 865 |
| PAC B-Type (E)C-(E)C | 10.28 | 577 |
| PAC B-Type (E)Cg-(E)GC | 13.1 | 745 |
| PAC B-Type (E)Cg-(E)GC | 14.35 | 897 |
| PAC B-Type (E)C-(E)GC | 20.07 | 593 |
| Dihydrochalcones | | |
| Phloretin-C-glucoside (nothofagin) | 12 | 435 |
| Phloretin-O-glucoside (phlorizin) | 13.23 | 435 |

**Table 3.** *Cont.*

| Stilbenes | | |
|---|---|---|
| Piceatannol-O-Glucoside (astringin) | 14.04 | 405 |
| **Acetophenones** | | |
| Myrciaphenone B | 12.68 | 481 |
| **Benzophenones** | | |
| Guavinoside A | 12.29 | 543 |
| Guavin B-isomer | 14.23 | 693 |
| Guavinoside B-isomer | 14.64 | 571 |
| Guavinoside B-isomer | 15.39 | 571 |
| Guavin B-isomer | 15.5 | 693 |
| Guavin B-isomer | 15.62 | 693 |
| Guavinoside B isomer | 15.82 | 571 |
| Glucopyranosyl-benzophenone | 18.12 | 557 |
| **Other polar compounds** | | |
| Cinnamoyl-hexoside | 9.9 | 355 |
| Abscisic acid-hexoside | 10.24 | 425 |
| Abscisic acid | 14.1 | 263 |
| **Anthocyanidins** | | |
| Cyanidin-3-O-glucoside | 6.51 | 449 |

| **Leaves and fruit [b]** | | |
|---|---|---|
| **Compound** | **Rt (min)** | **m/z** |
| **Benzophenone** | | |
| Guavinoside b isomer | 16.88 | 571.1431 |
| **Ellagic acid derivates** | | |
| Pedunculagin isomer | 6.63 | 783.0704 |
| Ellagic acid-O-pent oside | 11.39 | 433.0402 |
| **Flavan-3-ols** | | |
| (epi) catechin | 9.42 | 289.071 |
| **Flavonols** | | |
| Quercetin glucuronide | 12.38 | 477.0662 |
| Quercetin hexoside | 12.49 | 463.058 |
| Quercetin pentoside | 13.39 | 433.0779 |
| **Proanthocyanidins (PAC)** | | |
| PAC B-Type $C_{30}H_{26}O_{12}$ (E)C-(E)C | 6.98 | 577.1352 |
| PAC B-Type $C_{45}H_{38}O_{18}$ (E)C-(E)C-(E)C | 8.62 | 865.1977 |
| PAC B-Type $C_{45}H_{38}O_{18}$ (E)C-(E)C-(E)C | 10.06 | 865.1974 |

[a] Taken from: [50], [b] taken from: [55]. PAC B-Type: proanthocyanidin with a B-type linkage. Rt, retention time (min), m/z, mass-to-charge ratio.

*3.3. Seed*

The light yellow or cream seeds are flattened in a kidney shape, measuring between 3 and 5 mm long and 2 to 3 mm wide; the number of seeds per fruit can range between 100 and 500 [19,20], accounting for between 1.6 and 4% of the fruit's weight [19].

Guava seeds contain approximately 92% dry matter, of which 80% is fiber, 8 to 12.75% oil, 6 to 10% protein, and 0.5 to 6.62% ash [29,61]. According to Vasco-Méndez et al. [61], the fiber contains 25% lignin and 65% hemicellulose, and the ethereal extract contains the following fatty acids: 79% linolenic, 8% palmitic, 7% oleic, and 5% stearic. Triolein is the most abundant triglyceride (60%). According to Uchoa-thomaz et al. [24], the yield of guava seed meal is 54% when compared to other fruits in the same family, such as coronilla guava (*P. acetabulum*) with 55.01% and guava arazá (*Eugenia stipitata*) with 63%, and this has been attributed to the number of seeds present in *P. guajava* L.

According to chemical composition studies, the pulp, peel, and guava seed have pH values of 4.1, 3.9, and 4.30 ± 0.03, respectively. Because it is close to the value (4.5) that restricts microorganism growth, low pH values provide relative resistance to microbial attack [24,30]. The acidity of the fruit varies depending on factors such as variety

and maturity; studies report values ranging from $1.21 \pm 0.16$ to $2.18 \pm 0.08$ g of citric acid/100 g [24].

Pelegrini and Franco [34] discovered and characterized Pg-AMP1, an antimicrobial peptide that belongs to a group of glycine-rich proteins found in guava seeds and is distinguished by its low molecular weight and three-dimensional structure. Antimicrobial peptides from other families are similar. In vitro analysis revealed that it could inhibit the growth of gram-negative bacteria. In one of their studies, they discovered that at a concentration of 6.5 mM, *Proteus* and *Klebsiella* spp. growth was decreased by 30% and 90%, respectively. Pg-AMP1 has no activity against Gram-positive bacteria such as *Staphylococcus aureus*, indicating that it is only active against Gram-negative microorganisms. Serna-cock et al. [29] investigated guava seed flour as a nitrogen source in alcoholic fermentation and discovered increased product yield and substrate conversion. These findings support the use of guava seed flour as a sustainable and low-cost source of nitrogen for fermentation alcoholics.

Guava processing by-products, mostly seeds, peels, and pulp, have a water-holding capacity of 10.2 g water/g sample [34], which is greater than that reported for certain foods such as rice bran (5.21 g water/g sample) [62] and durum wheat (1.5–2.1 g water/g sample) [63], indicating its potential use in food production. However, some studies report that it has a negative impact when added to baked foods, resulting in a loss of volume and a sandy texture in the final product [62].

## 4. Anti-Nutritional Aspects

The term anti-nutritional refers to substances that modify normal elements of animal metabolism, affecting some food in a portion of its nutritional value, such as hindering or limiting the assimilation of some or several particular nutrients obtained from food [64,65]. They may also cause stomach bloating, gas accumulation in the digestive system, and even be poisonous, causing damage to organs such as the liver, pancreas, kidneys, or even blood disorders [42,64–66].

Condensed tannins, for example, are synthesized during seed development as a source of nutrition for the plant [35]. Proanthocyanidins are capable of causing food rejection in ruminant animals as well as reducing nutrient absorption, as are some catechins that interfere with mineral absorption, such as iron [60]. Sánchez-Zuiga et al. [67] conducted a study in which they discovered more than 60 compounds in guava puree. A compound (a hexoside Chrysin glucoside) from the Chrysin-C family of flavones was also discovered among them. It is found primarily in cereals and fruits like passion fruit, and it is used in dietary substitutes. It has, however, been linked to changes in male hormones when consumed in quantities greater than 4 g; in 2016, the Federal Food, Drug, and Cosmetic Act removed it from the list of substances approved for bulk use. Another compound found in this fruit is rutin, which was discovered in a variety of guava from Salvador Bahia, Brazil, by Santos et al. [37]. This is a compound with many described properties; however, it is not recommended for pregnant or lactating women, and it can cause photosensitivity in large amounts.

It is very common for these fruits to be transformed into food products such as juices. Rojas-Garbanzo et al. [51] identified some phytochemicals in Costa Rican guava juice belonging to several family groups of phenolic compounds. Despite the fact that this compound is found in many other fruits, including coffee, excessive consumption is not recommended for individuals with heart problems, and it is also known to have an effect on the nervous system; there have even been studies in which it has been identified as a possible carcinogen assessed in mice [68,69].

Various compounds have been observed on the leaves of this fruit over the years that may be considered antinutritional due to their ability to denature proteins that could affect animals and humans, such as the compounds of flavonoids and steroids families [35]. In addition to amine compounds and triterpenes such as saponins, the latter have surfactant properties that, when dissolved in water and stirred, may have detergent properties;

moreover, they may have hemolytic properties that affect the permeability of the biological membrane, making them poisonous to cold-blooded animals [70]. Other compounds, such as β-sitosterol and alkaloids, have been shown to be genotoxic in large quantities [71,72].

There have been no investigations or reports of adverse effects from guava seed consumption [1], but the fruit's activity as a cardiac depressant and blood sugar reducer has been recorded. As a result, people with heart disease or hypoglycemia should eat it in moderation [34]. More research into the inherent toxicity of triterpenes in seeds and leaves is needed, given that studies have shown that leaf oil is four times more potent than vincristine [1].

The chemical composition of the fruit varies depending on its maturity. An immature fruit has a higher activity of hydrolytic enzymes (α-amylase and β-amylase); a higher content of tannins, chlorophyll, cellulose, hemicellulose, and lignin; and a lower content of carotenoids [65]. The unripe fruit is not recommended for consumption because it is not digestible and can cause constipation, vomiting, and fever [1].

## 5. Future Trends and Research Opportunities

Guava is a product with numerous applications due to its functional biocomposites and previously discussed traditional uses that encourage health. Always guided by demand from actual human systems for the development of circular economies, the presence of potential compounds of interest of plants to benefit people creates an interesting field of study.

A high-impact issue in various fields that will contribute to the resolution of various nutritional, cosmetic, food, and environmental issues is the intention of creating value-added products or procedures that use underutilized biomass sources.

### 5.1. In Health and Cosmetic Fields

There is a great opportunity to make products with less synthetic ingredients, which benefits health, the economy, and the environment. For example, Dantas Mota et al. [73] developed a sunscreen formulation supplemented with methanolic extracts of guava pulp, where the extracts increased sunscreen in the formulation by allowing the reduction of different pharmaceutical photoprotective ingredients and producing a synergistic behavior with the other ingredients. The authors attributed the UV protection effect to components such as flavonoids and tannins. The authors also emphasized the low concentration of coumarins in the extract since these compounds have been associated with phototoxicity.

Another approach that could be used along the guava chain is the use of emerging technologies, such as that by Luo et al. [70], which extract polysaccharides from guava leaves using ultrasound. The authors also analyzed their antioxidant and antiglycemic effects, where the inhibitory effect against glucosidase was higher than the normally used drug control (acarbose), but when the profiles for the inhibitory effect in α-amylase were compared with the same control, it was lower. Nonetheless, the high agonist effect of acarbose may ameliorate some adverse effects, such as the accumulation of certain carbohydrates in the intestine and their subsequent fermentation by intestinal microbiota, resulting in gas formation; these findings suggest that these natural polysaccharides may be used as a complementary therapy in Diabetes mellitus with possibly fewer side effects.

### 5.2. In Food Field

Guava byproducts were demonstrated by Thu Thi Tran et al. [36], which could be used successfully in the food technology field. In the study of Thu Thi Tran et al. [36], antioxidants extracted from guava leaves were suggested for further incorporation into pork sausage. This intended to avoid the addition of synthetic antioxidants; the inclusion of antioxidants at 4000 ppm retards lipid oxidation and preserves the correct color attributes, being equal to the synthetic antioxidant typically used; the addition of this type of natural compound should decrease the potential health harm associated with the use of synthetic antioxidants. Furthermore, it is important to note that this research involved the incorpo-

ration of this antioxidant within a complex matrix, as well as the importance of avoiding the disruption of all quality and organoleptic parameters in the product. Other studies, such as that by Casarotti et al. [74], have shown that the addition of guava processing by-products can be used in other types of foods. In their study, Casarotti et al. [74] had used agro-waste created in guava processing industries to formulate a fermented milk-derived beverage. This resulted in a benefit complementation between fermented products and the remaining high-value products in industrial waste, intending to always produce novel products with potential health advantages, using low-cost ingredients and with intact organoleptic properties.

A study in which guava powder was added as a source of dietary fiber in a formulation of sheep-meat pellets discovered that the addition of guava powder in the formulation greatly reduced the pH value of the emulsion, giving the mixture higher stability and increasing the content of ash and moisture. Furthermore, the content of phenolic compounds and total dietary fiber was increased without substantially altering the product's organoleptic properties [28]. Besides, the addition of guava powder to the formulation was found to retard lipid oxidation of cooked meat pellets during refrigerated storage, indicating the high potential of this food as a source of antioxidant dietary fiber in meat foods, and this type of bioactive material has a high potential to be used in in vitro studies to stimulate the growth of probiotic bacteria [75]. Nobre et al. [33] conducted a study of supplementation in the diet of lambs with guava agro-industrial residues, in which they analyzed the animals' growth performance and discovered that supplementation with values higher than 30% of these residues reduced food intake, daily weight gain, and final weight. This may be due to a decrease in total digestible nutrient levels and an increase in lignin content provided by guava residues.

In terms of food quality improvement, Omitoyin et al. [76] suggested the incorporation of aqueous extracts from guava leaves into fish diets in aquaculture, enhancing their immune systems and lowering mortality rates.

### 5.3. In Bio-Remediation Field

An element that has received little attention is the incorporation of this material in the bioremediation process [77], which involves the generation of magnetic nanoparticles coupled to guava leaves for the generation of a strategy that permits the removal of methylene blue from water. The ground guava leaves signify support in this process that does not facilitate the release of dangerous substances into the water. Furthermore, the use of guava leaves is very significant in India because they represent a highly produced group of biomasses in one of the major guava-producing countries, producing an alternative for them. Other guava parts, such as seeds, are also used in the bioremediation processes which were suggested by Ramos Vargas et al. [78], in which they were modified with $AlCl_3$. Ramos Vargas et al. [79] investigated the effect of these hazardous compounds on the removal of fluoride and arsenate in aqueous solutions. The use of these materials as bio-absorbents produces a significant amount of waste that could be handled in ground waters with more effective and less expensive materials.

### 5.4. In Biotechnological Field

Another field that will benefit from the use of guava waste is biotechnology, as shown by Arshad et al. [80], who used guava seeds as support in solid-state fermentation to produce the enzyme, tannase. Biotechnology may also increase a wide range of functional properties by significantly increasing the release of phenols through solid-state fermentation by microorganisms such as Monascus anka and *Bacillus* spp. According to Wang et al. [78], biotechnology should be used to increase the potential of all agro-industrial wastes, including those associated with guava production and consumption.

### 5.5. In Other Fields

The scientific approach is always on the lookout for new uses for guava trash. For example, Araújo et al. [81] investigated the use of industrial waste from guava pulp as a cryoprotective agent for the preservation of various lactic acid bacteria (LAB), examining parameters such as survival rate, membrane damage and viability, and the protective effect of the guava. This type of bacteria should be used in food development or food technology research, where preservation is a critical process. Another novel application of guava is that suggested by Fitri et al. [82], who used guava to absorb fatty acids produced during the extraction or production of coconut oil. This fatty acid is quickly oxidized, resulting in rancid odors and a low-quality oil. Although this research used all of the fruit's material, it could be scaled down to use guava waste. Another demand that could be met using guava waste is the production of biofuels, as was proposed by Iha et al. [83], by first extracting the oil from the seed using soxhlet for a subsequent chemical modification to generate biodiesel and further thermal cracking at high temperatures to produce bio-oil, resulting in the production of biodiesel with the correct Brazilian specifications.

The entire guava contains highly valuable compounds that could be used to create goods and services that benefit human health. The current state and future trends seek a set of strategies such as incorporation in food, animal feed, cosmetics, pharmaceuticals, bio-absorbents, enzyme production, and biofuel production, often with a preference for green technologies; however, studies on new extraction technologies that allow them to retain their functionalities, as well as the study of all compounds at the in vivo level, are also important.

### 6. Conclusions

The phytochemical composition of both the fruit and the by-products of processing has revealed a valuable source of compounds present in considerable amounts, including dietary fiber (soluble and insoluble), vitamins (A, E, β-carotene), minerals (selenium, zinc), proteins (transferrin, ceruloplasmin, albumin), antioxidants, flavonoids, flavonols, and condensed tannins. Some of them are present in higher concentrations in by-products than in pulp; they force us to reconsider the current use of by-products waste. Several studies have demonstrated the numerous advantages and applications that can be offered, not only to fruit, but also to by-products, which can be applied directly or through the extraction of its components in the agricultural, pharmaceutical, biotechnological, bioremediation sectors, such as biofuels, and other fields that give residues a value equal to or greater than that of the fruit. It is important to highlight the numerous applications in the food industry and the studies that report the use of guava and its by-products as a low-cost functional ingredient that, when incorporated into formulations, improves the nutritional quality of food. Furthermore, the compounds found in the guava plant, fruit, and by-products have been shown to have a beneficial action in the body for the treatment of diseases.

One of the most significant opportunities provided by by-products is the use of dietary fiber, which is sought as an ingredient in the food industry, not only because of the amount present in the by-products, but also because of the balanced relationship between insoluble and soluble dietary fiber, given its significance for nutritional impacts. The by-products contain significant quantities of polyphenolic compounds with antioxidant capacities, which is why certain researchers regard these by-products to be potential nutraceutical resources. They are capable of being integrated into food formulations and increasing the quality and nutritional contribution of foods aimed at the low-income community. Others suggest using these to decrease the use of chemical preservatives in food production, and some studies emphasize their application in the formulation of meat sausages.

**Author Contributions:** Conceptualization, J.E.A.-L.; writing—original draft preparation, J.E.A.-L., C.T.-L., K.N.R.-G. and G.A.M.; writing—review and editing, J.E.A.-L., A.C.F.-G. and C.N.A.; supervision, A.C.F.-G. and C.N.A.; project administration, A.C.F.-G. and C.N.A. All authors have read and agreed to the published version of the manuscript.

**Funding:** This research received no external funding.

**Acknowledgments:** The authors are thankful to the National Council of Science and Technology (CONACYT) of Mexico and the postgraduate degree in Food Science and Technology of the Autonomous University of Coahuila. This financing was carried out by granting a postgraduate scholarship to the student, Jorge Eduardo Angulo López, with scholarship number/CVU 945769 for the completion of their master's studies. Authors thank the technical support given by Deepak Kumar Verma (Indian Institute of Technology–Kharagpur, India) to check the spelling and grammar of English language.

**Conflicts of Interest:** The authors declare no conflict of interest.

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
