# Peer review of "Guava (Psidium guajava L.) Fruit and Valorization of Industrialization By-Products"

_processes, doi:10.3390/pr9061075_

Round 1

Reviewer 1 Report

In this review, Angulo and colleagues carried out a review article on the application of guava and its by-products for industrial purposes. Despite the overall interest of this work, through reading of the abstract and particularly the whole manuscript, the authors failed in what aimed to do. The manuscript lacks of organization and scientific soundness, with most aspects being approached in a superficial manner besides that there is no specific data on guava by-products biovalorization. The section related to chemical composition of guava and its different parts should be completely restructured and the economic impact of this fruit should be clearly presented. Through reading of the current version of the manuscript, it seems that the authors initially aimed to focus on the biological potential of guava and then given the increasing demand by consumers for its use, the authors decided to move to its agro-industrial applications, but failed to achieve it.

Looking specifically at tables 1 and 2 most of information summarized in outdated and even some is repeated (as is the case of ref 17), in addition to that most information presented in the form of text is not summarized in the tables, along with the presence of contradictory data in text when looking at tables. In addition, and looking at table 3 the information should be presented by classes and subclasses of phenolic compounds.  

Author Response

Reviewers

Reviewer #1: 

In this review, Angulo and colleagues carried out a review article on the application of guava and its by-products for industrial purposes. Despite the overall interest of this work, through reading of the abstract and particularly the whole manuscript, the authors failed in what aimed to do. The manuscript lacks of organization and scientific soundness, with most aspects being approached in a superficial manner besides that there is no specific data on guava by-products biovalorization. The section related to chemical composition of guava and its different parts should be completely restructured and the economic impact of this fruit should be clearly presented. Through reading of the current version of the manuscript, it seems that the authors initially aimed to focus on the biological potential of guava and then given the increasing demand by consumers for its use, the authors decided to move to its agro-industrial applications, but failed to achieve it.

Looking specifically at tables 1 and 2 most of information summarized in outdated and even some is repeated (as is the case of ref 17), in addition to that most information presented in the form of text is not summarized in the tables, along with the presence of contradictory data in text when looking at tables. In addition, and looking at table 3 the information should be presented by classes and subclasses of phenolic compounds.  

R//. The evaluator's comment is very important to improve the quality of the manuscript. The change was performed according to the suggestion. The bioactive compounds were grouped by families according to the information reported in specialized scientific literature.

The information on the compounds present in the sheets was updated. A new scientific article was added (2021) in the references section:

Rojas-Garbanzo, C.; Rodríguez, L.; Pérez, A.M.; Mayorga-Gross, A.L.; Vásquez-Chaves, V.; Fuentes, E.; Palomo, I. Anti-platelet activity and chemical characterization by UPLC-DAD-ESI-QTOF-MS of the main polyphenols in extracts from Psidium leaves and fruits. Food Res. Int. 2021, 141, doi:10.1016/j.foodres.2020.110070.

Reviewer 2 Report

Authors describe in their review the physicochemical composition of the different by-products of the processing of guava, and their described uses. Although they compile part of the knowledge about the subject, during last year, 2020, have been published a high number of publications about it. As a review, probably to be published in 2021, it must be updated. For that reason, I recommend to accept the manuscript after a major revision.

In addition I have the following recommendations:

Lines 65-73, Information about other species of the genus Psidium, is it really needed?

Lines 104 and more, Please when including data of production, specify the years of the information.

Lines 78-101, When describing the tree, leaves and fruits... will be very helpful to include some photos.

Table 1, Please, create a new table just for guava flour data.

Line 173-175, Please express all the data in the same units, GAE/g or GAE/100g.

Line 249, “Table” term is missed.

Table 3. In “Leavesa” and “Fleshb”, final “a” and “b” as superscripts letters.

Line 273 Please, include reference.

Lines 261-275 Please check English.

Lines 276-343 Please check English.

Line 401 Authors mention the “multiples varieties” but no information about them are included in the review regards to the different content of interesting compounds.

References, What kind of sorting has been used?

Ref. 2, 4 and 10, They are book chapters, please, check journal references style.

Ref. 11 and 13, What kind of reference are they?

Ref. 26, 58 and 59, Check reference for completeness.

Ref 57, Add web link http://congresos.cio.mx/2_enc_mujer/Extenso/Posters/S1-QUI07.doc

Author Response

Comments and Suggestions for Authors

Authors describe in their review the physicochemical composition of the different by-products of the processing of guava, and their described uses. Although they compile part of the knowledge about the subject, during last year, 2020, have been published a high number of publications about it. As a review, probably to be published in 2021, it must be updated. For that reason, I recommend to accept the manuscript after a major revision.

In addition I have the following recommendations:

  • ✔ Lines 65-73, Information about other species of the genus Psidium, is it really needed?

R//. Yes, because they are the main varieties, the most consumed.

  • ✔ Lines 104 and more, Please when including data of production, specify the years of the information.

R//. …production of guava Psidium guajava L. is estimated to be approximately 40 million tons (2020)

… Mexico with a production of 302,718 tons/year (2017)

  • ✔ Lines 78-101, When describing the tree, leaves and fruits... will be very helpful to include some photos.

a)

b)

c)

d)

Figure 1: a) A guava plant, b) A flower of guava, c) Guava (Psidium guajava L.), d) Diversity in shapes and color among guava cultivars [2,14]

  • ✔ Line 173-175, Please express all the data in the same units, GAE/g or GAE/100g.

              The content of phenolic compounds in guava powder Psidium guajava L. is in the range of 55 - 516 mg GAE/100 g [33]. The pink pulp presents values between 170 to 300 mg GAE/100 g [23]. The main phenolic compounds found in the pulp are gallic acid, chlorogenic acid, ellagic acid, catechin and rutin [33]. The antioxidant activity of the extracts correlates with the variety of phenolic compounds found [2, 33].

  • ✔ Line 249, “Table” term is missed.

       R//. Term was added

  • ✔ Table 3. In “Leavesa” and “Fleshb”, final “a” and “b” as superscripts letters.

R//. Change made.

  • ✔ Line 273 Please, include reference.

R//. [56]

  • ✔ Lines 261-275 Please check English.

R//. reviewed and corrected

  • ✔ Lines 276-343 Please check English.

R//. reviewed and corrected

  • ✔ Line 401 Authors mention the “multiples varieties” but no information about them are included in the review regards to the different content of interesting compounds.

R//. A section of the paragraph was removed

References, What kind of sorting has been used?

R//. The Mendeley program was used in the Processes style.

  • ✔ 2, 4 and 10, They are book chapters, please, check journal references style.

R//.

  • (2) Gill, K.S. Guavas. In Encyclopedia of Food and Health; Caballero, B., Finglas, P.M. and Toldra, F., Eds.; Oxford: Academic Press. Punjab Agricultural University, Ludhiana, India, 2016; pp. 270-277. https:// doi.org/10.1016/B978-0-12-384947-2.00363-9.

  • (4) Hidalgo, F.R.; Gómez, U.M.; Escalera, C.D.; Quisbert, D.S. Beneficios de la guayaba para la salud. Rev. Inv. Inf. Salud [online] 2015, 10(25): 27-32. Disponible en: http://www.revistasbolivianas.org.bo/scielo.php?script=sci_arttext&pid=S2075-61942015000300005&lng=es.

  • (10) Irshad, Z.; Hanif, M.A.; Ayub, M.A.; Jilani, M.I.; Tavallali, V. Guava. In Medicinal Plants of South Asia, 1st ed.; Muhammad, A.H., Haq, N., Muhammad, M.K., Hugh, J.B., Eds.; Elsevier Ltd., United States of America, 2020; pp. 341–354. doi:10.1016/b978-0-08-102659-5.00026-4.

  • ✔ 11 and 13, What kind of reference are they?

R//. These references are thesis

  • (11) Solarte, M.E. Aspectos ecofisiológicos y compuestos bioactivos de guayaba (Psidium guajava L.) en la provincia de Vélez, Santander-Colombia, 2013. Doctoral thesis, Universidad Nacional de Colombia, Bogotá, 2013.

  • (13) Serrato, C. Efectos en la germinación de semillas de guayaba (Psidium guajava) consumidas por monos aulladores negros (Alouatta pigra) en Balancán, Tabasco, México., 2012. Undergraduate tesis, Benemérita Universidad Autónoma de Puebla, 2012.

  • ✔ 26, 58 and 59, Check reference for completeness.

R//.

  • (26) Mondragón, C; Toriz, LM; Guzmán, SH. Characterization of guava selection for the Bajio region of Guanajuato. Téc. Méx [online] 2009, vol.35, n.3, pp.315-322. ISSN 0568-2517.v

  • (58) Pérez, M.L.; Hernández, A.M.; Agroindustrial coproducts as sources of Novel Functional Ingredients; In Food Processing for Increased Quality and Consumption, 1st ed.; Grumezescu, A.M., Holban, A.M., Eds.; Elsevier Ltd., United States of America, 2018; pp. Pages 219-250. ISBN 9780128114476.

  • (59) Ciudad, M.; Fernández, V.; Matallana, M.C.; Morales, P. Dietary fiber sources and human benefits: The case study of cereal and pseudocereals. Adv Food Nutr Res. 2019, vol. 90, pp. 83-134. doi: 10.1016/bs.afnr.2019.02.002.

  • ✔ Ref 57, Add web link http://congresos.cio.mx/2_enc_mujer/Extenso/Posters/S1-QUI07.doc
  • ✔ R//. Copied

Reviewer 3 Report

Comments

In my opinion the manuscript meets the criteria for a review publication.

Minor remark:

Line 314: wáter – should be water

Table 3. Rt(min), m/z – explain what it means

Line 250: aTaken from: [46] and bTaken from: [47]  - no reference in the table

Line 60: P. guajava L. and Line 160: Psidium guajava L. – should be the same in the text.

Similary, Table 3, e.g. mg / 100 g and mg/ 100 g; Line 174: GAE/100 , Line 175: GAE / 100 g –  space or not, should be te same.

Author Response

Comments and Suggestions for Authors

Comments

In my opinion the manuscript meets the criteria for a review publication.

Minor remark:

  • ✔ Line 314: wáter – should be water

R//. The error is on line 213. the change was made

  • ✔ Table 3. Rt(min), m/z – explain what it means

Rt, retention time (min), m/z, mass-to-charge ratio.

 âœ”            Line 250: aTaken from: [46] and bTaken from: [47] - no reference in the table

R//. The reference is in the first row of the table, in each column.

 âœ”             

  • ✔ Line 60:  guajava L. and Line 160: Psidium guajava L. – should be the same in the text.

            R//. The terms were unified throughout the document.

  • ✔ Similary, Table 3, e.g. mg / 100 g and mg/ 100 g; Line 174: GAE/100, Line 175: GAE / 100 g – space or not, should be te same.

R//. Change made.

Round 2

Reviewer 1 Report

All comments were properly addressed, despite data organization and text fluency may also be improved. The most important aspect that attract the readers interest is the flow of ideas presented.

Author Response

We have attended all comments and suggestions

Reviewer 2 Report

Authors han improved the manuscript quality in this version. I only have some minor corrections/sugestions:

(i) Once introduced the first time the complete species name, please use the abreviate form "P. guajava".

(ii) Please, modify figure 1 legend... to something similar to: "Guava plant (a), flower (b), fruits (c) and diversity in shapes and color among cultivars."

Author Response

(i) Once introduced the first time the complete species name, please use the abreviate form "P. guajava".

R//. Change made.

(ii) Please, modify figure 1 legend... to something similar to: "Guava plant (a), flower (b), fruits (c) and diversity in shapes and color among cultivars."

R//.

Figure 1. Guava plant (P. guajava L.). a) Plant, b) flower, c) fruit, c) diversity of shapes and colors between cultivars
